# Factors Affecting Survival Outcomes in Neuroendocrine Tumor of the Appendix over the Past Two Decades

**DOI:** 10.3390/diseases12050096

**Published:** 2024-05-08

**Authors:** Vignesh Krishnan Nagesh, Izage Kianifar Aguilar, Daniel Elias, Charlene Mansour, Hadrian Hoang-Vu Tran, Ruchi Bhuju, Tanni Sethi, Paranjyothy Rao Pirangi Sanjeeva, Marco Gonzalez Rivas, Emelyn Martinez, Auda Auda, Nazir Ahmed, Shawn Philip, Simcha Weissman, John Sotiriadis, Ayrton Bangolo

**Affiliations:** 1Department of Internal Medicine, Hackensack Palisades Medical Center, North Bergen, NJ 07047, USA; izage.kianifaraguilar@hmhn.org (I.K.A.); de217@njms.rutgers.edu (D.E.); hadrian.tran@hmhn.org (H.H.-V.T.); emelyn.martinez@hmhn.org (E.M.); simcha.weissman@hmhn.org (S.W.); 2Department of Gastroenterology, Hackensack Palisades Medical Center, North Bergen, NJ 07047, USA; nazir.ahmed@hmhn.org (N.A.); shawn.philip@hmhn.org (S.P.); john.sotiriadis@hmhn.org (J.S.)

**Keywords:** neuroendocrine tumors, appendix, mortality, SEER database, prognostic factors

## Abstract

Background: Appendiceal neuroendocrine tumors (NETs) rank as the third most frequent neoplasm affecting the appendix, originating from enterochromaffin cells. This study aims to evaluate the influence of various prognostic factors on the mortality rates of patients diagnosed with NETs of the appendix. Methods: Conducted retrospectively, the study involved 3346 patients, utilizing data sourced from the Surveillance, Epidemiology, and End Results (SEER) database. Our analysis centered on investigating demographic characteristics, clinical features, overall mortality (OM), and cancer-specific mortality (CSM) among the cohort. Variables showing a *p*-value < 0.1 in the univariate Cox regression were incorporated into the multivariate Cox regression analysis. A Hazard Ratio (HR) > 1 indicated an unfavorable prognosis. Results: In the multivariate analysis, higher OM and CSM were observed in males, older age groups, tumors with distant metastasis, poorly differentiated tumors, and those who underwent chemotherapy. Non-Hispanic Black individuals showed elevated mortality rates. Conclusion: Delayed diagnosis may contribute to the increased mortality in this community. Improved access to healthcare and treatment is crucial for addressing these disparities. Larger prospective studies are needed to pinpoint the underlying causes of elevated mortality in non-Hispanic Black populations, and randomized controlled trials (RCTs) are warranted to evaluate therapies for advanced-stage appendix NETs.

## 1. Introduction

Appendiceal tumors are rare tumors, often diagnosed incidentally during appendectomies. These tumors are further classified as epithelial and non-epithelial tumors. The non-epithelial appendiceal tumors include lymphomas and appendiceal neuroendocrine tumors (aNETs) [1]. aNETs are the most common tumors of the appendix and account for 0.5–1% of intestinal tumors [2]. More than 80% of the cases of aNETs are diagnosed incidentally [3]. These tumors are island-like or tubular carcinoids that originate from neuroendocrine cells [1]. In recent decades, neuroendocrine tumors of the appendix have become more common due to improvements in the classification system, the presence of more developed imaging modalities, and a better understanding of the pathophysiology of these tumors [4]. The majority of the cases of aNETs are diagnosed in adults between the ages of 20–50 years [5]. These tumors are more common in women as compared to men, primarily due to a higher rate of incidental appendectomies in females who go through pelvic surgery [6]. The presence of aNETs does not present any symptoms; therefore, these tumors do not have a specific clinical description (5). Macroscopically, these tumors are whitish nodules and can be either irregular or well-demarcated in appearance [7]. After the incidental diagnosis of neuroendocrine tumors in the appendix, some additional tests are performed, which include endoscopy, biochemical tests, nuclear medicine imaging, and conventional imaging tests [8]. Circulating biomarkers are also used to diagnose aNETs; however, this method is not used for tumors that are unable to produce serotonin. This diagnostic procedure also does not work for patients who do not have carcinoid features [9].

The most promising treatment for a neuroendocrine tumor of the appendix is surgery. The surgical procedure can involve simple appendectomy or right hemicolectomy (RHC). After the incidental discovery of a neuroendocrine tumor in the appendix, the next step is to decide whether appendectomy is sufficient or right hemicolectomy is required to dissect the lymph node to prevent the chances of reoccurrence [10,11]. Several factors are considered before the treatment of a neuroendocrine tumor of the appendix. The size of the tumor is an important factor in deciding the extent of surgery. The size at which more complicated surgery is required is still controversial [12]. aNETs smaller than 20 mm can be eliminated by simple appendectomy and have less chance of recurrence and metastasis. If the tumor is larger than 20 mm, more aggressive interventions are required for its treatment [1]. Lympho-vascular invasion is another significant factor for the treatment of aNETs. It is an independent risk factor for the involvement of lymph nodes at right hemicolectomy [13]. Samples should be prepared carefully because certain artifacts can contribute to the overestimation of lympho-vascular invasion during analysis [14]. The location of the tumor is another parameter to decide the extent of surgery for patients with appendiceal neuroendocrine tumors [15]. Liver-directed therapy is another treatment option for metastatic aNETs. The dominant metastasis sites for aNETs are the liver and regional lymph nodes. Liver-directed therapy is recommended for patients in which tumors are metastatic to the liver and surgical procedures are not feasible. However, prospective and retrospective trials for this therapy are very limited [16,17,18]. A novel therapy to treat aNETs is radiolabeled somatostatin analog therapy [19]. Other therapies include targeted therapies and cytotoxic chemotherapy [20,21]. The prognosis of aNETs is generally good, and the survival rate is 5 years in more than 90% of the cases [1]. Follow-up depends on the type of surgery and the features of the tumor. Although mortality is low for aNETs, they impact the quality of life of the patients. Therefore, proper management is crucial for the patients with aNETs [12].

Currently, there is a paucity of data regarding factors that determine the survival outcomes in patients suffering from aNETs. Therefore, the present study aimed to explore the factors that affect the survival outcomes in the patients who have aNETs over the past two decades.

## 2. Materials and Methods

A retrospective cohort study of patients with neuroendocrine tumors of the appendix was carried out using the Surveillance, Epidemiology, and End Results (SEER) database, which includes data from 18 registries and the November 2020 submission (http://www.seer.cancer.gov), accessed on 9 March 2024. The study comprised data on patients from the years 2000–2017. The SEER database is supported by the United States National Cancer Institute (US NCI). The SEER Program, known for being one of the most comprehensive and credible cancer data sources in the United States, compiles the SEER 18 database. This database records cancer incidence, as well as clinical and pathological details of patients and survival outcomes from 18 population-based cancer registries, encompassing roughly 28% of the US population. Data from the SEER database were retrieved using histological codes for the diagnosis of neuroendocrine tumor (NET) along with the primary site code corresponding to the appendix. Patients with an unspecified age at diagnosis, race, or stage of neuroendocrine tumor (NET) of the appendix development were excluded from the study. In this study, all the variables considered were treated as the primary exposures.

Overall mortality (OM) refers to deaths from any cause by the end of this study period. Cancer-specific mortality (CSM) is defined as deaths resulting from complications associated with neuroendocrine tumors (NETs) of the appendix by the study’s conclusion. Variables extracted for the study included age at diagnosis, gender, race (White, Black, and others), ethnicity (non-Hispanic and Hispanic), tumor grade, stage at diagnosis (localized, regional, distant), residential geographic area, annual income, marital status, year of diagnosis, and treatment methods, such as surgery, radiation, and chemotherapy.

The Cox proportional hazards regression model presupposes that hazard rates stay consistent over time. In the analysis, variables showing a *p*-value below 0.1 in the univariate Cox regression were selected for inclusion in the multivariate Cox proportional hazards model. This approach was used to identify independent predictors of overall mortality (OM) and cancer-specific mortality (CSM), where a hazard ratio (HR) exceeding 1 indicates detrimental prognostic factors. All evaluations were two-tailed, employing a 95% confidence interval, and findings with a *p*-value under 0.01 were considered to be statistically significant. These statistical assessments were conducted using the STATA 18 software.

## 3. Results

This study analyzed 3346 patients diagnosed with a neuroendocrine tumor of the appendix between 2000 to 2017. The demographic and clinicopathologic characteristics, as outlined in Table 1, indicate that females (59.38%), individuals aged between 40–59 years (37.84%), non-Hispanic Whites (76.39%), residents in counties with a population exceeding 1 million (59.29%), and those with an annual income of USD 75,000 or higher (42.50%) were predominantly represented in the cohort. The distribution across age groups showed 34.13% in the 0–39 years category, 37.84% in 40–59 years, 24.99% in 60–79 years, and 3.05% in the 80+ years category. Marital status analysis revealed that the majority were married (50.66%), followed by single (36.31%), divorced/separated (8.88%), and widowed (4.15%). Analysis of tumor stage distribution revealed a significant proportion of cases in the localized stage (71.58%), with other stages comprising regional (20.92%) and distant (7.50%) cases. Tumor subtypes showed that the majority of tumors were well differentiated (grade 1) (72.95%), followed by moderately differentiated (grade 2) (15.96%), poorly differentiated (grade 3) (9.59%), and undifferentiated (grade 4) (1.49%). The racial composition was led by non-Hispanic Whites (76.39%), followed by Hispanics (11.86%), non-Hispanic Blacks (7.56%), and other (4.18%). Treatment patterns indicated that 90.62% did not undergo chemotherapy and 99.94% underwent surgery. The incidence of new cases diagnosed each year ranged from 0.36% in 2000 to 22.27% in 2017.

Table 2 presents the crude analysis of several factors associated with all-cause mortality and cancer-related mortality among US patients diagnosed with a neuroendocrine tumor of the appendix between 2000 and 2017. The assessment of overall mortality and cancer-related mortality is conducted across various characteristics. Noteworthy findings include age playing a critical role, with the 40–59 age group (HR = 7.73, 95% CI 5.07–11.79, *p* < 0.01), the 60–79 age group (HR = 15.02, 95% CI: 9.87–22.86, *p* < 0.01), and the 80+ age group (HR = 34.45, 95% CI 21.20–55.98, *p* < 0.01) exhibiting increased risks. Marital status displayed a decrease in HR for those single (HR = 0.47 95% CI 0.36–0.63, *p* < 0.01). Among tumor stages, tumors with regional spread had a higher mortality (HR = 1.79, 95% CI: 1.43–2.23–6.56, *p* < 0.01). The diversity within tumor subtypes exerted a considerable influence on overall mortality, with noteworthy distinctions observed among various grades. Specifically, moderately differentiated tumors (grade 2) exhibited a discernible impact (HR = 2.20, 95% CI 1.74–2.79, *p* < 0.01), while this effect was markedly accentuated for poorly differentiated (grade 3) tumors (HR = 13.28, 95% CI 10.89–16.19, *p* < 0.01) and undifferentiated (grade 4) tumors (HR = 5.56, 95% CI 3.54–8.72, *p* < 0.01). Additionally, the Hispanic race had a lower OM (HR = 0.63, 95% CI 0.45–0.89). Notably, individuals subjected to chemotherapy displayed increased risks (HR = 1.53, 95% CI 1.17–2.01, *p* < 0.01). Income per year and living area did not show significant associations.

For cancer-related mortality, age played a crucial role, with the 40–59 age group (HR = 8.54, 95% CI: 5.10–14.30, *p* < 0.01), the 60–79 age group (HR = 12.22, 95% CI 7.27–20.54, *p* < 0.01), and 80+ age group (HR = 17.90, 95% CI: 9.13–35.12, *p* < 0.01) exhibiting increased risks. Tumor stage showed an increased risk of cancer-related mortality for the regional (HR = 3.79, 95% CI 2.72–5.28, *p* < 0.01) and distant stages (HR = 39.94, 95% CI 29.94–53.28, *p* < 0.01). Tumor subtypes were a significant factor for increased risk, specifically moderately differentiated (grade 2) (HR = 3.39, 95% CI 2.40–4.77, *p* < 0.01), poorly differentiated (grade 3) (HR = 21.42, 95% CI 16.29–28.18, *p* < 0.01), and undifferentiated (grade 4) (HR = 14.07, 95% CI 8.50–23.39, *p* < 0.01) tumors. Individuals who were Hispanic had a lower CSM (HR = 0.57, 95% CI 0.36–0.91, *p* < 0.05). Remarkably, individuals undergoing chemotherapy (HR = 13.10, 95% CI 10.49–16.35, *p* < 0.01) exhibited considerable susceptibility to cancer-related mortality. Living area and income per year did not show significant associations.

Table 3 displays the outcomes of multivariate Cox proportional hazard regression analyses investigating factors influencing overall mortality and cancer-related mortality among patients in the United States diagnosed with a neuroendocrine tumor of the appendix from 2000 to 2017. In terms of overall mortality, males exhibited a heightened risk with an adjusted proportional hazard ratio of 1.26 (95% CI 1.05–1.52, *p* < 0.05) compared to females. Noteworthy findings include age playing a critical role, with the 40–59 age group (HR = 4.67, 95% CI 2.98–7.33, *p* < 0.01), the 60–79 age group (HR = 10.47, 95% CI: 6.68–16.42, *p* < 0.01), and the 80+ age group (HR = 29.02, 95% CI 16.96–49.66, *p* < 0.01) exhibiting increased risks. Marital status displayed a progressive increase in HR for those single (HR = 1.33, 95% CI 1.06–1.68, *p* < 0.05), divorced/separated (HR = 1.39, 95% CI 1.05–1.85, *p* < 0.05), and widowed (HR = 1.60, 95% CI 1.13–2.25, *p* < 0.01). Among tumor stages, only the distant stage had a significant impact on overall mortality (HR = 4.93, 95% CI 3.70–6.56, *p* < 0.01). The diversity within tumor subtypes exerted a considerable influence on overall mortality, with noteworthy distinctions observed among various grades. Specifically, moderately differentiated tumors (grade 2) exhibited a discernible impact (HR = 1.31, 95% CI 1.02–1.68, *p* < 0.05), while this effect was markedly accentuated for poorly differentiated (grade 3) tumors (HR = 2.91, 95% CI 2.24–3.78, *p* < 0.01) and undifferentiated (grade 4) tumors (HR = 2.06, 95% CI 1.26–3.39, *p* < 0.01). Additionally, race and living area emerged as significant factors influencing overall mortality, as evidenced by the elevated hazard ratios among non-Hispanic black individuals (HR = 1.41, 95% CI 1.04–1.92, *p* < 0.05) and those residing in metropolitan areas with populations exceeding 250,000 (HR = 1.51, 95% CI 1.05–2.16, *p* < 0.05). Notably, individuals subjected to chemotherapy displayed increased risks (HR = 1.53, 95% CI 1.17–2.01, *p* < 0.01). Income per year did not show significant associations.

For cancer-related mortality, male gender remained a significant risk factor (HR = 1.32, 95% CI 1.03–1.68, *p* < 0.05). Age played a crucial role, with the 40–59 age group (HR = 3.22, 95% CI 1.83–5.67, *p* < 0.01), the 60–79 age group (HR = 5.61, 95% CI 3.17–9.92, *p* < 0.01), and 80+ age group (HR = 12.51, 95% CI 5.91–26.51 *p* < 0.01) exhibiting increased risks. Tumor stage showed an increased risk of cancer-related mortality for the regional (HR = 2.00, 95% CI 1.40–2.85, *p* < 0.01) and distant stages (HR = 11.89, 95% CI 8.07–17.51, *p* < 0.01). Tumor stages were a significant factor for increased risk, specifically moderately differentiated (grade 2) (HR = 1.74, 95% CI 1.21–2.50, *p* < 0.01), poorly differentiated (grade 3) (HR = 4.23, 95% CI 2.96–6.05, *p* < 0.01), and undifferentiated (grade 4) (HR = 3.39, 95% CI 1.92–5.98, *p* < 0.01) tumors. Individuals who were non-Hispanic Black remained a risk factor (HR = 1.55, 95% CI 1.05–2.29, *p* < 0.05). Remarkably, individuals undergoing chemotherapy (HR = 1.56, 95% CI 1.13–2.15, *p* < 0.01) exhibited considerable susceptibility to cancer-related mortality. Marital status, living area, and income per year did not show significant associations. Overall, the multivariate analyses demonstrate how various factors impact both overall and cancer-related mortality among patients with a neuroendocrine tumor of the appendix. This underscores the significance of considering multiple variables in understanding survival outcomes in these patients. The age-adjusted mortality rate was 0.1 per 100,000 of the United States (US) population, a very low rate, indicating few deaths relative to the size of the US population due to the rarity of the cancer.

## 4. Discussion

The study analyzed 3346 US patients diagnosed with neuroendocrine tumors of the appendix between 2000 and 2017. Demographic analysis revealed a predominance of females, individuals aged 40–59, non-Hispanic Whites, residents in populous counties, and those with higher incomes. Most tumors were well-differentiated and at localized stages. Crude and multivariate analyses identified several significant factors affecting overall and cancer-related mortality. We discovered significant associations indicating a poor prognosis for patients with poorly differentiated tumors, advanced age, and advanced tumor stages. Specifically, poorly differentiated tumors exhibited substantially higher hazard ratios for both overall mortality and cancer-related mortality, suggesting a more aggressive disease course. Additionally, advanced age groups, particularly those aged 60 and above, demonstrated increased risks for mortality outcomes. Moreover, we observed higher OM and CSM rates among non-Hispanic Black individuals compared to other racial groups, indicating racial disparities in survival outcomes.

The available literature on NET, particularly those of the gastrointestinal (GI) tract and appendix, remains sparse, with limited long-term follow-up data and survival analyses. Epidemiological information, clinical presentations, and natural history of NET are largely derived from registry databases, lacking detailed clinical insights. A retrospective survey conducted across 13 Italian referral centers enrolled 820 patients with thoracic, gastroenteropancreatic (GEP), or metastatic NET of unknown primary origin (U-NET). Results showed a linear increase in NET incidences from 1990 to 2007, with pancreas and lung being the most common primary sites. The mean age at diagnosis was 60 years, though significantly earlier in patients with multiple endocrine neoplasia type 1 (MEN1). Symptoms varied, with tumor burden being the most common presentation, followed by endocrine syndromes and fortuitous diagnoses. Hormonal hypersecretions, notably insulin and serotonin, were prevalent. Advanced tumor stages were more frequent in GI and thymic NET [22].

Another study aimed to identify prognostic factors influencing survival after pancreatectomy for pancreatic neuroendocrine tumors (PNETs) and develop a post-resection prognostic score. Analyzing data from the National Cancer Data Base (1985–2004), encompassing 3851 patients who underwent PNET resection, the study found that age, tumor grade, presence of distant metastases, tumor functionality, and type of resection were independent predictors of survival. Notably, gender, race, socioeconomic status, tumor size, nodal status, margins, adjuvant chemotherapy, and hospital volume did not significantly impact survival. Age, grade, and distant metastases emerged as the most significant predictors and were integrated into the prognostic score, which demonstrated strong correlations with outcomes and offered valuable survival discrimination [23].

In the study by Lewkowicz et al. (2015) focusing on gastroenteropancreatic neuroendocrine neoplasms (GEP-NENs), clinical characteristics and factors influencing 5-year survival were investigated in 122 patients diagnosed between 2002 and 2011 in Kraków or its administrative region. The most common primary tumor sites were the small intestine, pancreas, rectum, stomach, appendix, and colon. Tumor differentiation varied, with most cases being well-differentiated. Notably, higher tumor grade (NEN G2), advanced stage according to the AJCC/UICC classification, and the presence of metastases at diagnosis were associated with poorer prognosis in univariate analysis. However, in multivariate analysis, advanced stage and the presence of metastases at diagnosis emerged as independent risk factors for death. Overall, 5-year survival was 85%, highlighting the importance of early detection and aggressive management in patients with GEP-NENs, particularly those presenting with advanced stage or metastases. This single-center study provides valuable insights for identifying patients with poorer prognoses who may benefit from a more intensive treatment approach [24].

Garcia-Carbonero et al. (2010) conducted a study utilizing data from a National Cancer Registry to explore GEP-NENs in Spain. The group studied included 907 tumors, largely comprising carcinoids, pancreatic nonfunctional tumors, metastatic NETs with no identifiable primary origin, insulinomas, and gastrinomas. Significant variations in the stage at diagnosis were observed and were influenced by factors such as gender, the location of the primary tumor, tumor type, and grade. The overall five-year survival rate stood at 75.4%, with more favorable outcomes in women, younger individuals, patients exhibiting hormonal syndromes, and those with tumors that were either early-stage or of a lower grade. Importantly, the stage and Ki-67 index were the sole independent variables found to predict survival, emphasizing their critical role in prognosis evaluation [25].

The study by Shen et al. focused on examining racial disparities in the incidence and survival rates among patients with neuroendocrine tumors (NETs). Their analysis revealed that Black individuals exhibited higher incidence rates of NETs across all stages, with the greatest disparity observed in the local stage compared to Whites. Moreover, despite exhibiting clinical characteristics generally linked to a better prognosis, Black patients with advanced-stage NETs had poorer survival outcomes. This research underscored substantial negative differences in socioeconomic and sociodemographic factors among Black patients, indicating that social determinants, support systems, and healthcare accessibility might play a role in the noted disparities in NET incidence and survival rates across different racial groups [26].

Contrary to previous findings, Goksu et al. conducted a study utilizing the SEER database to investigate the impact of race and ethnicity on disease characteristics and survival outcomes in gastrointestinal NETs, including appendiceal NETs. Analyzing data from 26,399 patients diagnosed between 2004 and 2015, they found significant differences in patient demographics and tumor characteristics among racial and ethnic groups. In particular, non-Hispanic White patients tended to be male, over the age of 60, and more frequently diagnosed with metastatic disease compared to Hispanic and non-Hispanic Black patients. Notably, Hispanic patients demonstrated better overall survival rates, whereas non-Hispanic Black patients showed improved cause-specific survival rates when compared to their non-Hispanic White counterparts. These findings were confirmed in multivariable analysis, indicating that race and ethnicity serve as independent prognostic factors in patients with gastrointestinal NETs [27].

In our research, the mortality rate was higher among patients diagnosed with appendix neuroendocrine tumors (NETs) who underwent chemotherapy compared to those who did not, likely due to chemotherapy administration to patients with metastatic disease [2]. However, small-scale studies focusing on non-pancreatic NET patients treated with dacarbazine chemotherapy did not demonstrate improved survival outcomes [28,29]. Limited data exist regarding chemotherapy effectiveness in appendix NETs, although multiple clinical trials are investigating mTOR inhibitors, tyrosine kinase inhibitors, and angiogenesis inhibitors for metastatic NETs [20]. Everolimus, an mTOR inhibitor, has been evaluated for gastropancreatic NETs [30], while sunitinib, a tyrosine kinase inhibitor targeting VEGF, has been approved for metastatic pancreatic cancer treatment [2]. Surufatinib, an oral multikinase inhibitor inhibiting VEGF, is in phase III clinical trials for extrapancreatic NET treatment [31].

Our study revealed that nearly all patients underwent surgical intervention. According to the National Comprehensive Cancer Network (NCCN), simple appendicectomy is recommended as the primary treatment for localized appendix NETs less than 2 cm [32,33]. Tumors sized between 1.0 and 1.9 cm have a higher likelihood of nodal metastasis; however, studies indicate that right hemicolectomy does not confer benefits for these patients, with such intervention reserved for tumors exceeding 2 cm in size [33]. Surgical excision for metastatic NET tumors may still be considered, particularly for patients with symptomatic liver metastases, aiming to alleviate symptoms [16]. Newer guidelines show that simple appendicectomy would suffice for tumors less than the size of 1 cm and right hemicolectomy may be warranted for tumors > 2 cm. However, surgery for tumors 1–2 cm is still controversial, especially tumors with high-risk features [2].

Our findings suggest that poorly differentiated tumors are associated with increased mortality due to their aggressive nature and tendency to present with metastatic disease at diagnosis [7,34].

Our study was conducted on a large sample size of 3346 patients with tissue diagnosis of NET of the appendix. The patient population was selected with strict inclusion and exclusion criteria. However, our studies have weaknesses as it is a retrospective study, and the SEER database does not take into account other comorbidities or the type of surgery or chemotherapy the patient received. Furthermore, reasons for delayed treatment and/or diagnosis are not provided.

## 5. Conclusions

In conclusion, our study’s findings demonstrate the impact of several factors, including race, age, tumor stage, and treatment modalities, on the survival outcomes in patients with NETs of the appendix. We found that people diagnosed at older ages, those receiving chemotherapy, and patients with advanced grades, distant stages, or poorly differentiated NETs have a higher risk of both OM and CSM. In terms of contributing factors, males and non-Hispanic Black individuals exhibited higher mortality rates, highlighting potential disparities in care. Given that these tumors are rare and often found as incidental findings, our results may represent a delay in diagnosis and should prompt the medical practice to be more familiar with the pathophysiology/clinical manifestations of these tumors as a way to increase the rate of diagnosis and, thus, improve patient outcomes and prognosis. These findings align with the existing literature on NETs by reinforcing the understanding that factors such as age at diagnosis, tumor grade, and stage significantly impact patient outcomes. This comprehensive study on neuroendocrine tumors (NETs) of the appendix, alongside auxiliary research, provides critical insights into refining precision medicine, enhancing medical quality, and ensuring patient safety, which are pivotal in the context of healthcare provision. By advocating for personalized treatment plans tailored to tumor specifics and patient demographics and promoting the use of genetic profiling, healthcare providers can significantly improve the prognosis for patients with NETs. Previous studies have also noted the increased mortality risk associated with advanced grades and stages of NETs, as well as the potential benefits of personalized treatment approaches. This study also points to the need for further research to elucidate the underlying causes of racial disparities and to identify optimal treatment strategies for different patient subgroups. Genetic factors may predispose non-Hispanic Black communities and males to have higher mortality, as seen in our study; however, larger prospective studies to identify the causes of this correlation are needed. The information provided in our study can be used as a point of reference to design further investigations, including but not limited to randomized controlled trials (RCTs) needed to enlighten treatment options or guidelines to redirect the approach and management of NETs. Our study paves the way for physicians to identify these ethnic groups who have higher mortality from NETs of the appendix and provide them with earlier screening and treatment to improve survival outcomes.

## Figures and Tables

**Table 1 diseases-12-00096-t001:** Demographic and clinicopathologic characteristics of US patients diagnosed with neuroendocrine tumor of the appendix between 2000 and 2017.

Characteristics		
**Total**	**N**	**%**
	**3346**	**100%**
**Gender**		
Female	1987	59.38
Male	1359	40.62
**Age at diagnosis, y.o**		
00–39	1142	34.13
40–59	1266	37.84
60–79	836	24.99
80+	102	3.05
**Marital status**		
Married	1695	50.66
Single	1215	36.31
Divorced/separated	297	8.88
Widowed	139	4.15
**Tumor stage**		
Localized	2395	71.58
Regional	700	20.92
Distant	251	7.50
**Tumor subtypes**		
Well differentiated (Grade 1)	2441	72.95
Moderately differentiated (Grade 2)	534	15.96
Poorly differentiated (Grade 3)	321	9.59
Undifferentiated (Grade 4)	50	1.49
**Race**		
Non-Hispanic White	2556	76.39
Non-Hispanic Black	253	7.56
Hispanic	397	11.86
Other	140	4.18
**Living area**		
Counties in metropolitan areas of 1 million persons	1984	59.29
Counties in metropolitan areas of 250,000 to 1 million persons	739	22.09
Counties in metropolitan areas of 250,000 persons	242	7.23
Nonmetropolitan counties adjacent to a metropolitan area	217	6.49
Nonmetropolitan counties not adjacent to a metropolitan area	164	4.90
**Income per year**		
<USD 35,000	30	0.90
USD 35,000–44,999	145	4.33
USD 45,000–54,999	300	8.97
USD 55,000–64,999	576	17.21
USD 65,000–74,999	873	26.09
USD 75,000+	1422	42.50
**Chemotherapy**		
No	3032	90.62
Yes	314	9.38
**Surgery**		
No	2	0.06
Yes	3344	99.94
**Year of diagnosis**		
2000	12	0.36
2001	12	0.36
2002	6	0.18
2003	15	0.45
2004	18	0.54
2005	25	0.75
2006	21	0.63
2007	28	0.84
2008	40	1.20
2009	41	1.23
2010	113	3.38
2011	131	3.92
2012	221	6.60
2013	256	7.65
2014	359	10.73
2015	618	18.47
2016	685	20.47
2017	745	22.27

**Table 2 diseases-12-00096-t002:** Crude analysis of factors associated with all-cause mortality and cancer-related mortality among US patients diagnosed with neuroendocrine tumor of the appendix between 2000 and 2017.

Characteristics	Overall Mortality. Adjusted Proportional Hazard Ratio (95% Confidence Interval)	Cancer-Related Mortality. Adjusted Proportional Hazard Ratio (95% Confidence Interval)
**Gender**		
Female	1 (reference)	1 (reference)
Male	1.15 (0.97–1.36)	1.15 (0.92–1.44)
**Age at diagnosis, y.o**		
00–39	1 (reference)	1 (reference)
40–59	7.73 (5.07–11.79) **	8.54 (5.10–14.30) **
60–79	15.02 (9.87–22.86) **	12.22 (7.27–20.54) **
80+	34.45 (21.20–55.98) **	17.90 (9.13–35.12) **
**Marital status**		
Married	1 (reference)	1 (reference)
Single	0.47 (0.36–0.63) **	0.47 (0.36–0.63) **
Divorced/separated	1.12 (0.79–1.59)	1.12 (0.79–1.59)
Widowed	1.45 (0.92–2.30)	1.45 (0.92–2.30)
**Tumor stage**		
Localized	1 (reference)	1 (reference)
Regional	1.79 (1.43–2.23) **	3.79 (2.72–5.28) **
Distant	13.28 (10.89–16.19) **	39.94 (29.94–53.28) **
**Tumor subtype**		
Well differentiated (Grade1)	1 (reference)	1 (reference)
Moderately differentiated (Grade 2)	2.20 (1.74–2.79) **	3.39 (2.40–4.77) **
Poorly Differentiated (Grade 3)	13.28 (10.89–16.19) **	21.42 (16.29–28.18) **
Undifferentiated (Grade 4)	5.56 (3.54–8.72) **	14.07 (8.50–23.39) **
**Race**		
Non-Hispanic White	1 (reference)	1 (reference)
Non-Hispanic Black	1.27 (0.95–1.70)	1.34 (0.93–1.92)
Hispanic	0.63 (0.45–0.89) **	0.57 (0.36–0.91) *
Other	0.83 (0.51–1.34)	0.96 (0.54–1.71)
**Living area**		
Counties in metropolitan areas of 1 million persons	1 (reference)	1 (reference)
Counties in metropolitan areas of 250,000 to 1 million persons	0.92 (0.74–1.15)	0.90 (0.68–1.19)
Counties in metropolitan areas of 250,000 persons	1.15 (0.84–1.58)	0.77 (0.48–1.25)
Nonmetropolitan counties adjacent to a metropolitan area	0.92 (0.63–1.34)	0.68 (0.39–1.16)
Nonmetropolitan counties not adjacent to a metropolitan area	1.22 (0.86–1.73)	1.41 (0.93–2.15)
**Income per year**		
<USD 35,000	1 (reference)	1 (reference)
USD 35,000–44,999	1.08 (0.41–2.82)	1.45 (0.33–6.40)
USD 45,000–54,999	1.02 (0.41–2.55)	1.32 (0.31–5.55)
USD 55,000–64,999	1.10 (0.45–2.69)	1.74 (0.43–7.09)
USD 65,000–74,999	0.89 (0.36–2.17)	1.44 (0.35–5.84)
USD 75,000+	0.86 (0.36–2.10)	1.33 (0.33–5.37)
**Chemotherapy**		
No	1 (reference)	1 (reference)
Yes	6.57 (5.49–7.87) **	13.10 (10.49–16.35) **

* *p* < 0.05, ** *p* < 0.01.

**Table 3 diseases-12-00096-t003:** Multivariate Cox proportional hazard regression analyses of factors affecting all-cause mortality and cancer related mortality among US patients diagnosed with Neuroendocrine tumor of the appendix between 2000 and 2017.

Characteristics	Overall Mortality. Crude Proportional Hazard Ratio (95% Confidence Interval)	Cancer Related Mortality. Crude Proportional Hazard Ratio (95% Confidence Interval)
**Gender**		
Female	1 (reference)	1 (reference)
Male	1.26 (1.05–1.52) *	1.32 (1.03–1.68) *
**Age at diagnosis, y.o**		
00–39	1 (reference)	1 (reference)
40–59	4.67 (2.98–7.33) **	3.22 (1.83–5.67) **
60–79	10.47 (6.68–16.42) **	5.61 (3.17–9.92) **
80+	29.02 (16.96–49.66) **	12.51 (5.91–26.51) **
**Marital status**		
Married	1 (reference)	1 (reference)
Single	1.33 (1.06–1.68) *	1.31 (0.97–1.77)
Divorced/separated	1.39 (1.05–1.85) *	1.14 (0.79–1.66)
Widowed	1.60 (1.13–2.25) **	1.45 (0.86–2.44)
**Tumor stage**		
Localized	1 (reference)	1 (reference)
Regional	1.07 (0.84–1.37)	2.00 (1.40–2.85) **
Distant	4.93 (3.70–6.56) **	11.89 (8.07–17.51) **
**Tumor subtype**		
Well Differentiated (Grade 1)	1 (reference)	1 (reference)
Moderately Differentiated (Grade 2)	1.31 (1.02–1.68) *	1.74 (1.21–2.50) **
Poorly Differentiated (Grade 3)	2.91 (2.24–3.78) **	4.23 (2.96–6.05) **
Undifferentiated (Grade 4)	2.06 (1.26–3.39) **	3.39 (1.92–5.98) **
Non-Hispanic white	1 (reference)	1 (reference)
Non-Hispanic black	1.41 (1.04–1.92) *	1.55 (1.05–2.29) *
Hispanic	1.33 (0.94–1.89)	1.32 (0.82–2.12)
Other	0.90 (0.54–1.48)	0.86 (0.46–1.59)
**Living area**		
Counties in metropolitan areas of 1 million persons	1 (reference)	1 (reference)
Counties in metropolitan areas of 250,000 to 1 million persons	0.87 (0.69–1.09)	0.83 (0.61–1.12)
Counties in metropolitan areas of 250,000 persons	1.51 (1.05–2.16) *	1.14 (0.67–1.94)
Nonmetropolitan counties adjacent to a metropolitan area	0.88 (0.55–1.38)	0.71 (0.37–1.36)
Nonmetropolitan counties not adjacent to a metropolitan area	0.96 (0.61–1.51)	1.16 (0.67–2.00)
**Income per year**		
<$35,000	1 (reference)	1 (reference)
$35,000–44,999	0.98 (0.37–2.61)	1.65 (0.36–7.54)
$45,000–54,999	0.99 (0.38–2.56)	1.59 (0.36–7.06)
$55,000–64,999	0.94 (0.36–2.45)	1.81 (0.41–8.02)
$65,000–74,999	0.71 (0.27–1.90)	1.28 (0.28–5.77)
$75,000+	0.81 (0.30–2.13)	1.39 (0.31–6.26)
**Chemotherapy**		
No	1 (reference)	1 (reference)
Yes	1.53 (1.17–2.01) **	1.56 (1.13–2.15) **

* *p* < 0.05, ** *p* < 0.01.

## Data Availability

The data used and/or analyzed in this study are available in the Surveillance, Epidemiology, and End Results (SEER) Database of the National Cancer Institute (http://seer.cancer.gov), accessed on 9 March 2024.

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
