# Peer review of "Factors Affecting Survival Outcomes in Neuroendocrine Tumor of the Appendix over the Past Two Decades"

_diseases, 2024, doi:10.3390/diseases12050096_

Round 1

Reviewer 1 Report

Comments and Suggestions for Authors

This is a manuscript looking into factors that affects mortality rate of patients with NET in appendix.

Abstract- clear

Introduction- Clear. Just a small comment on line 64- to remain the consistency, please change ANETs to aNETs.

Methods- Clear inclusion and exclusion criteria.

1.     Please expand on SEER in line 80.

2.     Could the authors explain the significance of ‘November 2020 submission’?

3.     Timeline of the included patients from database should be stated in methods section instead of results section.

Results-

1.     As the results are detailed in Table 1, the 1st paragraph of results section is repeating what has been shown in the table. The authors could reduce what is written in the 1st paragraph.

2.     Could the authors include some details on the 2 patients who had no surgery. Did they have chemotherapy or palliated?

3.     The hazard ratio results for tumour stage for both Overall Mortality and Cancer Related Mortality are the same. Could the authors check if this is correct?

4.     Could the authors include the mortality rate of this cohort?

Discussion- The references used for treatment guidelines are from 1987/ 2017 and are not specific to aNET. There are newer guidelines eg: DOI 10.3390/cancers15010295

Author Response

This is a manuscript looking into factors that affects mortality rate of patients with NET in appendix.

Re: Thank you very much for taking the time out of your busy schedule to review our paper. Important and relevant points noted. 

Abstract- clear

Introduction- Clear. Just a small comment on line 64- to remain the consistency, please change ANETs to aNETs.

Re: Thank you for this keen observation ,changes have been made to the manuscript

Methods- Clear inclusion and exclusion criteria.

  1.     Please expand on SEER in line 80.

      Re: Thank you for this keen observation, changes have been made and highlighted in red.

  1.     Could the authors explain the significance of ‘November 2020 submission’?

       Re: This means that the database used was last updated in November 2020. 

  1.     Timeline of the included patients from the database should be stated in methods section instead of results section.

       Re: Thank you for this keen observation, changes have been made to the manuscript. 

Results-

  1.     As the results are detailed in Table 1, the 1st paragraph of results section is repeating what has been shown in the table. The authors could reduce what is written in the 1st paragraph.

      Re: Thank you for this keen observation , repetitions have been deleted and the paragraph was made more concise.

  1.     Could the authors include some details on the 2 patients who had no surgery. Did they have chemotherapy or palliated?

       Re: Thank you for this keen observation , unfortunately due to the granularity of the SEER database, the details on the therapy for the patients who did not receive the surgery were not provided.

  1.     The hazard ratio results for tumour stage for both Overall Mortality and Cancer Related Mortality are the same. Could the authors check if this is correct?

       Re: Thank you for this keen observation, it was an error and the table has been updated

  1.     Could the authors include the mortality rate of this cohort?

     Re: The age-adjusted mortality rate was 0.1 per 100,000 of the United States (US) population, a very low rate, indicating few deaths relative to the size of the US population due to the rarity of the cancer. And this was added in the Results section as well. Highlighted in red. 

Discussion- The references used for treatment guidelines are from 1987/ 2017 and are not specific to aNET. There are newer guidelines eg: DOI 10.3390/cancers15010295

Re: Thank you for this keen observation newer data has been added to the discussion.

Reviewer 2 Report

Comments and Suggestions for Authors

This paper by Nagesh et al., entitled “Factors affecting Survival outcomes in Neuroendocrine Tumor of the appendix over the past 2 decades” regards a large retrospective study involving 3346 patients, using data from the Surveillance, Epidemiology, and End Results (SEER) database. Authors concluded that delayed diagnosis may contribute to the increased mortality in this community, so suggesting that improved access to healthcare and treatment is crucial for addressing these disparities.

Specific Comments

1.     Limitations should include retrospective design.

2.     In the title, the number 2 should run as two.

3.     In Table 1 statistical differences, where appropriate, should be added. This would further strenghten gender, tumor subtype, tumor stage, race, living area, treatment modality and year of diagnosis differences.

Author Response

This paper by Nagesh et al., entitled “Factors affecting Survival outcomes in Neuroendocrine Tumor of the appendix over the past 2 decades” regards a large retrospective study involving 3346 patients, using data from the Surveillance, Epidemiology, and End Results (SEER) database. Authors concluded that delayed diagnosis may contribute to the increased mortality in this community, so suggesting that improved access to healthcare and treatment is crucial for addressing these disparities.

Re: Thank you very much for taking the time out of your busy schedule to review our paper. Important and relevant points noted. 

Specific Comments

  1.     Limitations should include retrospective design.

     Re:  Thank you for this keen observation , changes have been made and highlighted in red

  1.     In the title, the number 2 should run as two.

   Re: Thank you for this keen observation , changes have been made to the title and highlighted in red

  1.     In Table 1 statistical differences, where appropriate, should be added. This would further strengthen gender, tumor subtype, tumor stage, race, living area, treatment modality and year of diagnosis differences.

Re: Thank you for the comment but no analysis was run in Table 1. Table 1 was used to summarize the baseline characteristics. However, all statistically significant data have been added to table 2 where it was missing. 

Reviewer 3 Report

Comments and Suggestions for Authors

Reviewer’s comments to Author:

1.    Could the authors include in the analysis reasons for delayed diagnosis? Is it relevant to local healthcare policies? How to remind or advise residents about healthcare and education policies regarding healthcare in the future?

2.    This study cannot show whether there are differences in the relevant clinical symptoms of the study subjects? Is there any correlation in family history? However, it can be discussed by reviewing the information in the reference literature.

3.   This study cannot show whether the preoperative diagnosis and postoperative pathological diagnosis are consistent.

4.   There is no relevant data that analyzes whether the surgical methods and treatment methods are meaningful factors, parameters or indicators that affect postoperative treatment and survival rate. It is impossible to distinguish whether there are differences or correlations. Can you briefly summarize the discussion by citing reviewed literature?

5.    The tumor staging in Table 2 and 3  should be analyzed based on the AJCC eighth edition TNM staging.

6.   What suggestions does this study have for precision medicine, medical quality and patient safety? What are the learning objectives?

7.    Please briefly describe the strengths, weaknesses and limitations in your approach to this case. What difficulties need to be overcome?

Author Response

Thank you very much for taking the time out of your busy schedule to review our paper. Important and relevant points noted. 

  1.   Could the authors include in the analysis reasons for delayed diagnosis? Is it relevant to local healthcare policies? How to remind or advise residents about healthcare and education policies regarding healthcare in the future?

Re: Thank you for this insightful comment, However, reasons for delay in treatment are not provided in the SEER database used for this study. And this was added in the weakness of the study. It is relevant for healthcare policies and our study paves the way for future survey-based retrospective and prospective studies  that can focus on those reasons to improve the management of this rare malignancy. 

  1.   This study cannot show whether there are differences in the relevant clinical symptoms of the study subjects? Is there any correlation in family history? However, it can be discussed by reviewing the information in the reference literature.

    Re: The SEER database use for this study is ICD-10/ICD-03  based and does not provide information about family history of patients. Furthermore, the authors fail to see the relevance of such information.        

  1.   This study cannot show whether the preoperative diagnosis and postoperative pathological diagnosis are consistent.

    Re: Thank you for this keen observation , The SEER database use for this study is ICD-10/ICD-03  based, thus we are confident that preoperative diagnosis and postoperative pathological diagnosis are consistent. 

  1.   There is no relevant data that analyzes whether the surgical methods and treatment methods are meaningful factors, parameters or indicators that affect postoperative treatment and survival rate. It is impossible to distinguish whether there are differences or correlations. Can you briefly summarize the discussion by citing reviewed literature?

       Re: Thank you for this keen observation , changes have been made to the discussion section based on the literature (line 318, reference 37). 

  1.   The tumor staging in Table 2 and 3  should be analyzed based on the AJCC eighth edition TNM staging.

        Re: Thank you for this keen observation , however , the data was retrieved on patients from the years 2000-2017 , TNM data available in the SEER database is from 1988 to 2003. Thus, the staging system used for this study is appropriate. 

  1.   What suggestions does this study have for precision medicine, medical quality and patient safety? What are the learning objectives?

       Re: The study highlights the importance of early detection of NET appendix as earlier diagnosis as better prognosis. There is a higher mortality rate in NH black which could be due to disparities in access to healthcare or genomic factors, for which larger trials should be conducted. This has been highlighted in the conclusion section. 

  1.   Please briefly describe the strengths, weaknesses and limitations in your approach to this case. What difficulties need to be overcome

      Re: Thank you for considering our study, the strengths , weaknesses, and limitations have been mentioned in the last paragraph of the discussion section

Round 2

Reviewer 3 Report

Comments and Suggestions for Authors

The author has provided reasonable, scientific and logical responses to the reviewer's suggestions. I have no comments.